# Learning State Switching for Improved Exploration in Multi-sensor Environments

**Homagni Saha** [1]  **Sin Yong Tan** [1]  **Zhanhong Jiang** [2]  **Soumik Sarkar** [1]

## Abstract

Reinforcement learning has been used to achieve state of the art results in several robotics applications. Despite massive success, issues remain regarding transfer to real world domains and being able to optimize over multi-dimensional control inputs as well as across several agents with different perspective views for the same environment, but sharing the same overall goal (multi-agent, multi-sensor robotics). This paper takes a fresh approach towards multi-sensor multi-agent integration for achieving improved performance using a control theoretic approach of "state-switching". A formulation based on state switching is adapted as a multi-agent reinforcement learning task in the form of a value iteration algorithm maximizing expected payoffs over time. A reinforcement learning task involving tracking an unknown object with unknown motion dynamics using manipulators is formulated for the well known sawyer one handed manipulator. Thereafter we formulate our state switching algorithm and show superior performance compared to using individual sensors. Our trained agent is then transferred from simulation to a real setup and is shown to perform nicely in the real domain as well.

## 1. INTRODUCTION

There exist several practical applications that can benefit from consideration through a switching system perspective. The controls community is rich in literature pertaining to optimal scheduling and control of switching systems, with applications in robotics (Axelsson et al., 2005a), aerospace and mechanical systems (Rinehart et al., 2008; Heydari & Balakrishnan, 2013a), bio-engineering (Heydari & Balakrishnan, 2013b), and chemical processes (Liu & Gong, 2014).

[1]Department of Mechanical Engineering, Iowa State University, Ames, IA 50011, USA. [2]Johnson Controls International, 507 East Michigan St, Milwaukee, WI 53202, USA. Correspondence to: Soumik Sarkar <soumiks@iastate.edu>.

Planning and control in such scenarios is challenging because it has to deal with stabilization of a system composed of subsystems with different dynamics as well as deciding when to switch to a particular mode. Solutions to these types of control problem are predominantly based on nonlinear programming (Xu & Antsaklis, 2002; 2004; Axelsson et al., 2008; Ding et al., 2009; Axelsson et al., 2005b; Kamgarpour & Tomlin, 2012; Wardi & Egerstedt, 2012), and those that solve it as a search problem over a set of discretized switching instants (Luus & Chen, 2003),(Rungger & Stursberg, 2011). Recently Adaptive Dynamic Programming (ADP) based approaches have also been used successfully for control of switching systems in (Heydari & Balakrishnan, 2014). We employ state switching in the system in our consideration to achieve improved control. Active exploration for robot grasping using trajectory optimization has been done in (Kahn et al., 2015). In this paper we look at state switching with the benefits of improved exploration and robust control. The same system can be made to act out different control policies based on switching modes of control. We formulate an algorithm that employs this kind of deliberate switching using reinforcement learning based deep convolutional Q networks. The algorithm can be applied in any scenario involving multiple sensor based control. We demonstrate and validate our algorithm in the case of a 7 degree of freedom manipulator using multiple cameras that are able to capture multi-angle views of an object of interest for tracking an object for pick and place task. Multi-camera based 3 dimensional control of robots has been studied in (González-Galván et al., 2003; Wang et al., 2018; Ebert et al., 2018). Many consider some model based assumptions, whereas our approach is still model free. In this work we try to provide a new approach to the problem as well as propose a generic algorithm for multi-sensory control of systems in the perspective of a switching system. We formulate our problem as a reference tracking problem where the system is able to choose between operating modes. There exist reinforcement learning framework for these kinds of problems as in (He & Jagannathan, 2005). However our approach is novel in the sense that while modeling the system it considers the optimal switching sequence between several operational modes as well as the optimal control input once in a certain operational mode. This is achieved by coupling together optimal control and optimal switching decisions in

our algorithm.
**Contributions**

1. For the first time to the best of our knowledge investigate the effectiveness of simultaneous planning, control and scheduling using reinforcement learning approach.

2. Provide an algorithm for control based on multi-sensor input and validation results on a real robotic arm.

The remaining sections are organized as follows. In section 2 we combine state switching in the value iteration approach. Based on a 7 DoF robotic manipulator with raw image as inputs, in section 3 we go through deep Q learning in brief and provide our algorithm for visuomotor control. In section 4 we provide our experimental results and show improved performance both in simulation and experiments.

## 2. Control using state switching- Model, assumptions and goals

We consider the problem of controlling the states of a system to track a certain time varying signal. The decision variables are jointly comprised of the active mode in the given switching system as well as the optimal control input for the given active mode. In the context of controlling the joints of a manipulator we define different modes of the system as the different states in which the agent collects information about the environment. We model the discreet time control affine system for the scenario as follows where $x_k$ is our input state at time instant $k$:

$$x_{k+1} = f_i(x_k) + g_i(x_k)u_i(k), \ k \in \mathbb{N}, i \in \mathcal{I} := \{1, 2, ...M\} \tag{1}$$

Here $M$ is used to denote the maximum number of switching or active modes in the system, $f_i : \mathbb{R}^n \longrightarrow \mathbb{R}^n$ is continuous mapping that transforms the state $x$ at time instant $k$ based on the drift dynamics of the active mode $i$, $u_i(k) \in \mathbb{R}^m$ is the control input at time instant $k$ for the active mode $i$, and $g_i : \mathbb{R}^n \longrightarrow \mathbb{R}^{n \times m}$ is the input dynamics for the mode $i$ which is coupled with the $m$ dimensional control input. The goal in our problem can be defined as determining a switching schedule $i_k \forall k \in 0, 1, ..., \infty$, as well as determining the optimal control input $u_i(k)$ for the system at time instant $k$ and mode $i$, so that we could track a reference signal $v_k \in \mathbb{R}^s$ with an unknown dynamics $F : \mathbb{R}^s \longrightarrow \mathbb{R}^s$ as given by the equation below :

$$v_{k+1} = F(v_k) \tag{2}$$

Given an initial condition $v_0 \in \mathbb{R}^s$ traditional control theoretic methods formulate the solution as minimizing the cost function $J$ as given below:

$$J = \sum_{k=0}^{\infty} \mathcal{C}(x_k, v_k) \tag{3}$$

The standard assumption (Heydari, 2015) is that $\mathcal{C} : \mathbb{R}^n \times \mathbb{R}^s \longrightarrow \mathbb{R}^+$ is convex, continuous and positive semi-definite. Here, $\mathbb{R}^+$ is used to denote the set of non-negative reals. In the context of reinforcement learning we will reformulate minimization of cost function $J$ as maximization of payoff function $R$ as:

$$R = \sum_{k=0}^{\infty} \mathbb{P}(x_k, v_k) \tag{4}$$

Now, $\mathbb{P} : \mathbb{R}^n \times \mathbb{R}^s \longrightarrow \mathbb{R}^+$ provides a measure of how close to the goal the agent gets to after taking a certain action. The authors in (Heydari, 2015) propose a value iteration algorithm for the case when $x_{k+1} = f_i(x_k)$ (interested only in switching decisions) and establish convergence and feasibility of using Neural Networks (NN) as universal function approximators. In this section we will extend this technique over cases involving simultaneous control and switching decisions as modelled in equation 1. A typical value iteration approach would consider assigning a value to a particular configuration of the current state and the target tracking state. The optimal value for a certain pair of state $x_k$ and tracking reference $v_k$ can be formulated as a maximization of an infinite sequence, and equivalently as a recursion as follows :

$$V^*(x_k, v_k) = \mathbb{P}(x_k, v_k) + \sum_{j=k+1}^{\infty} \mathbb{P}(x_j^*, v_j) \tag{5a}$$

$$V^*(x, v) = \mathbb{P}(x, v) + V^*(f_{i*}(x) + g_{i*}(x)u_{i*}^*(x), F(v)) \tag{5b}$$

Here $x_j^* \forall j \in \{k+1, k+2, ...\}$ are the optimal future states calculated using the system dynamics, $i_j^* \in \mathcal{I}, \forall j \in \{k, k+1, ...\}$ are the optimal switching decisions, and $u_{i*}^* \in \mathbb{R}^m$ denote the optimal control input once in a certain active mode $i^*$. For practical purposes we can consider a parameter $w$ discretizing each dimension of our $m$ dimensional control input into a set of inputs of size $w$ representing our *domains of interest*. For example, for $m = 3$ and $w = 3$, we can have $u = [c_1, c_2, c_3]^T, c_1, c_2, c_3 \in \{1, 2, 3\}$, where 1,2 or 3 may directly map to certain value of a control input, based on designers knowledge of the operating characteristics of the system. Let $\mathbb{C} = \mathcal{P}(u)$ denote the power set of $u$, denoting the set of all possible control input vectors. Also, Let us denote $\mathcal{P}(i, \mathbb{C})$ as the *action set* $\mathcal{A}$, i.e. the set of all possible tuples of $i$, the active mode and $\mathbb{C}$, the corresponding control. In that case we can let :

$$\tau_a(x) = f_i(x) + g_i(x)u_i^j(x), u^j \in \mathcal{P}(\mathbb{C}), a \in \mathcal{P}(i, \mathbb{C}) \tag{6}$$

Based on the formulation in equation 6, we can write the update law of the simplest value iteration method which iteratively updates the value function as :

$$V^{j+1}(x, v) = \mathbb{P}(x, v) + \max_{a \in \mathcal{A}} V^j(\tau_a(x), F(v)) \tag{7}$$

The tuple $(x, v)$ in equation 7 can be thought of as an "augmented state" consisting of the current image captured by the camera which may or may not contain relevant information about the position of the object as well as the current joint angles of the robot. In reinforcement learning setting, the value function $V$ can be estimated by so called Q function parameterized by the current augmented state, action and weights of the neural network. Including $\tau_a$ in the formulation of value function inherently adds a "planning program objective" to the Q function and distinguishes it from a purely "reactive style" planner. This concept of embedding planners in RL agents to achieve improvements has been also recently considered in value iteration networks (Tamar et al., 2016).

## 3. Deep Reinforcement learning for visuomotor control

### 3.1. System Description

In this work we control the seven degree of freedom one handed manipulator -sawyer to be able to move its end effector as close as possible to a 3 dimensional object moving randomly on a 2 dimensional surface. A schematic of the Sawyer robot showing the degrees of freedom and joint assignment along with the placement of camera sensors is shown in figure 1. This will be necessary for describing the states of the system. The robot is connected to local server using Robot Operating System (ROS) and is able to transmit captured data to the local workstation at 100 Hz. In this experiment we try to learn a data driven model for the controller that takes in as input the raw images from two cameras along with the current values of the joint angles of the robot and output a desired change in joint angle values of the seven joints each time step and a desired mode of operation of the robot. We classify the operation of the robot into two distinct modes:

1. The state input to the controller comprises of images captured from the hand camera and the current joint angle values of the seven joints.

2. The state input to the controller comprises of images captured form the head camera and the current joint angle values of the seven joints.

The motivation for this state switching approach is primarily three reasons:

1. The high number of links in this type of serial link manipulator leads to a number of cases where the head camera cannot see the object to track.

2. We want to explore the feasibility of a multi-camera approach in place of a depth scan cameras that provide direct estimation of object locations, an approach

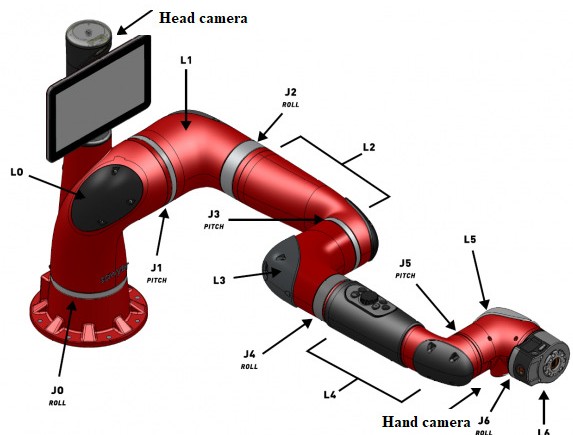

*Figure 1. Description of the joints*

commonly used in these kinds of tasks (Krainin et al., 2010). The switching between head and hand cameras is expected to provide comparable tracking results without any explicit camera calibration.

A forward kinematics calculation provides the Cartesian coordinates of the end point of the end effector of the robot at each time instant. During optimization of the neural network weights in the controller, it is possible to find the exact Euclidean distance of the end point of the end effector to the center of the 3 dimensional object for a set of fixed locations of the 3 dimensional object. This distance is used to calculate the *extrinsic-reward* to the reinforcement learning algorithm. In this framework we use two kinds of rewards for an agent: "extrinsic-reward", and "intrinsic-reward" both of which would be described in later sections. We first train our algorithm on Gazebo simulator provided for sawyer robot with a setup for a pick and place task. Later we transfer the trained neural network weights to fine-tune on the real world application. Figure 2 shows the simulator and the real world setup respectively.

### 3.2. Deep reinforcement learning framework

Deep Q Networks (DQN) is a model free implementation of value iteration method using Bellman equation (Dolcetta & Ishii, 1984). It is generally formulated as an iterative model-free solution to a Markov Decision Process (MDP) (Howard, 1960). An MDP is defined as a tuple $(S, A, P, r, \gamma)$ where:

1. S denotes the system state space. In our case, the state consists of the image captured by the robot camera ($\mathbb{I}$) concatenated with the current joint angle values ($\mathbb{J}$). Thus $S = \mathbb{I} \oplus \mathbb{J}$

2. A is the set of actions that can be taken by an agent. In our case $A = \mathcal{A}$. Each joint of the robot can either move up, move down or stay in the same position.

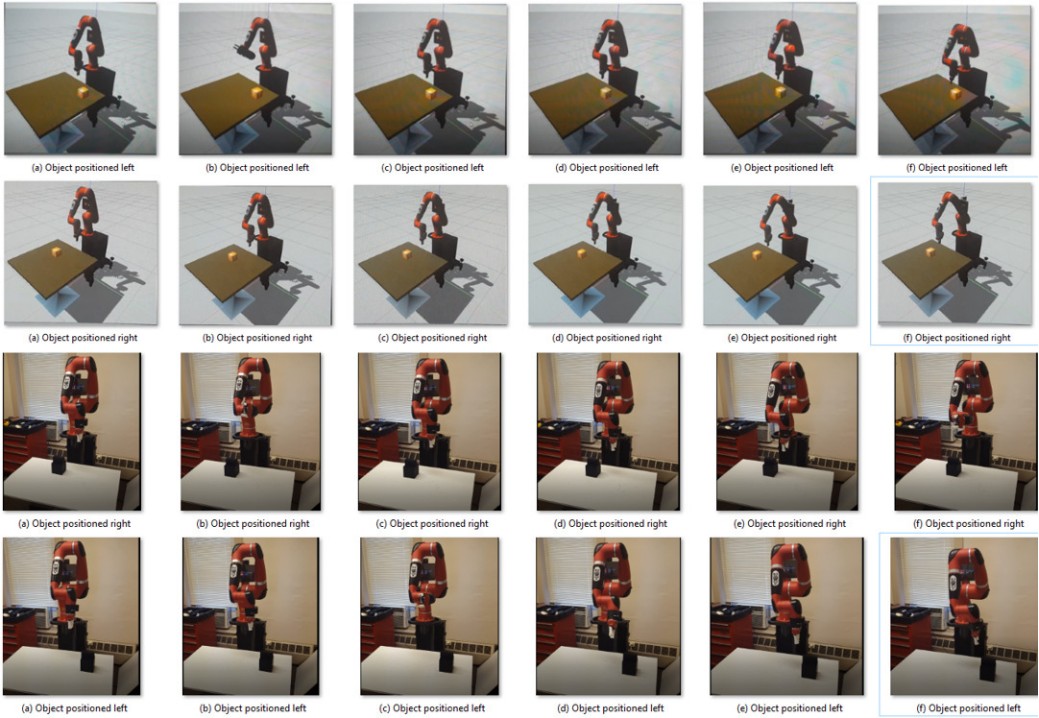

*Figure 2. Simulation and real experiments*

Provided there are 7 joints in the robot, and we are considering state switching between 2 cameras, $|\mathcal{A}| = 2 * (3^7)$.

3. P is the transition probability matrix that stores the probability of transitioning from a state $s_1 \in S$ to a state $s_2 \in S$ on taking an action $a \in A$.

4. $r(s, a)$ is a reward function that assigns the utility of taking an action $a$ in the state $s$. It is also parameterized by the current mode of the system $m$ and each joint angle input of each mode. Here we design two different kinds of reward function:

   (a) **Extrinsic-reward**: We denote it by $r^{ext}$. In our case, it is the exact decrease in Euclidean distance from the end point of the end effector to the center of the object to track due to action taken at a time instant. The reward function is available only during training and is supplied corresponding to control input for each of the joints when in a certain mode. However we expect our algorithm to generalize in testing.

   (b) **Intrinsic-reward**: We denote it by $r^{int}$. This reward is evaluated as the difference between entropy of Q values predicted by the network to the maximum possible entropy of the Q values. Details are provided in the next section.

5. $\gamma \in [0, 1]$ is the discount factor, a controlled parameter to assign greater relative importance to actions taken in the current step than those taken in the past. In our case we use a discount factor of 0.9.

Our algorithm takes an action just before time instant $k$ and receives rewards just after time instant $k$ based on the movement of the arm at time step $k$. For our experiments we restrict extrinsic reward function so that $r^{ext}(s, a) \in [-1, 1]$. This is achieved naturally by restricting the joint increment angles for individual joints. In the above section we mentioned that this was directly related to the Euclidean distance during training, thus physically, our reward varies from -1 meter to 1 meter. By formulation, intrinsic reward $r^{int}(s, a) \in [0, 1]$. Also we consider a fixed time horizon $K$ over which to optimize, essentially the number of trials the multi agent system is provided to move the robot as close as possible to the object, before being reset to the initial conditions. Being model free, our algorithm can also be applied to MDPs with unknown transition probabilities. Essentially for each mode $m$, it tries to maximize the following function which is known as the expected discounted total return from a starting time step $k_0$:

$$R^m(k_0) = \mathbb{E}\left( \sum_{k=k_0}^{K-1} (\gamma^{k-k_0} r^m(k)) \right) \forall m = \{1, 2, ...\} \quad (8)$$

Now $Q(x_k, a_k)$ will denote the utility of applying action $a_k$ in state $x_k$. Let $\pi$ denote a set of mappings from all states to actions. In that case, the optimal utility of the state action

pair $(x_k, a_k)$ would be given by:

$$Q^*(x, a) = \max_\pi \mathbb{E}[R_t | x_k = x, a_k = a, \pi] \qquad (9)$$

This would translate to solving the following Bellman equation :

$$Q^*(x, a) = \mathbb{E}_{\tilde{x} \sim \varepsilon}[r + \gamma \max_{\tilde{a} \in \mathcal{A}} Q^*(\tilde{x}, \tilde{a}) | x, a] \qquad (10)$$

Being model free, here $\tilde{x} \sim \varepsilon$ implies sampling states from unknown system dynamics $\varepsilon$. Since we are using deep convolutional neural networks for approximation of $Q$, we will need to parameterize this $Q$ function in terms of the weights of the neural network $\phi$. After successful training of the neural network, we will have :

$$Q(x, a, \phi) = Q^*(x, a) \qquad (11)$$

In order to achieve this, we would use the time varying loss function on the *behaviour distribution* of the agent $\rho(x, a)$ (Mnih et al., 2013), as described in equation 12 and apply minibatch gradient descent on a batch of data sampled from observed history by the agent.

$$L_k(\phi_k) = \frac{1}{2} \mathbb{E}_{x, a \sim \rho(x, a)}[(Q(x, a, \phi_k) - y_k)^2] \qquad (12a)$$

$$y_k := \mathbb{E}_{\tilde{x} \sim \varepsilon}[r + \gamma \max_{\tilde{a}} Q(\tilde{x}, \tilde{a}, \phi_{k-1}) | x, a] \qquad (12b)$$

We store this data in a memory buffer and in DQN terms this buffer is called the *Experience replay buffer*. We use $B_m$ to denote the buffer for a certain mode and $\mathcal{D}$ to denote the sampled batch size from each of these memory buffers. Being an online algorithm, control (switching and tracking) and optimization of the weights are done in an interleaved fashion. An epsilon greedy approach (Wunder et al., 2010) is also followed while picking actions to enable greater exploration in earlier stages. Below we provide our entire algorithm for end to end training of the controller.

---

**Algorithm 1** State switching DQN

**Result:** Set of neural network weight matrices $\Phi = \{\phi_1, ..., \phi_m\}$ for $m$ modes

Initialize the set of weight matrices $\Phi$

Initialize the set of target weight matrices $\Phi^t = \{\phi_1^t, ..., \phi_m^t\}$ for $m$ modes

Initialize $|m|$ replay buffers $\mathcal{B} = B_1, B_2, ... B_m$

**while** *time step $k \leq K$* **do**

 Let $\phi_c$ be the neural network weights for the current mode.

 With probability $\epsilon$ select a random action $a_k \in \mathcal{A}$

 With probability $1 - \epsilon$ select action $a_k = \text{argmax}_{a \in \mathcal{A}}(Q(x_k, a, \phi_c))$

 Let $u_k^j \subset a_k$ be the control input at the current state, execute $u_k^j$ on the system, observe extrinsic reward $r_k^{ext}$ and next state $x_{k+1}$

 Let $u_k^m \subset a_k$ be the decision for the next mode at the current state, observe intrinsic reward $r_k^{int}$, change the mode of the system accordingly to $c_{next}$ applying switching $u_k^m$ on $x_k$.

 Store the tuple $(x_k, a_k, r_k, x_{k+1})$ in the replay memory $B_c$

 **for** $B_i \in \mathcal{B}$ **do**

  **if** *size of replay buffer $B_i$ increased by $\mathcal{D}$* **then**

   Sample random minibatch transitions $(x_j, a_j, r_j, x_{j+1})$ from $B_i$

   Corresponding to $(x_j, a_j, r_j, x_{j+1})$ set:

   **if** *mode of the system was changed from $i$ to $\tilde{i}$* **then**

   $\quad | \quad y_j = r_j + \gamma \max_{\tilde{a}}(Q(x_{j+1}, \tilde{a}, \phi_{\tilde{i}}^t))$

   **end**

   **if** *mode of the system was unchanged* **then**

   $\quad | \quad y_j = r_j + \gamma \max_{\tilde{a}}(Q(x_{j+1}, \tilde{a}, \phi_i^t))$

   **end**

   Update $\phi_i$ using loss function given by : $\mathcal{L} = (y_j - Q(x_j, \tilde{a}, \phi_i))^2$

   Slightly push target weights towards actual weights : $\phi_i^t = 0.1 \times \phi_i$

  **end**

 **end**

**end**

---

### 3.3. Evaluating intrinsic switching reward

The Shannon entropy quantifies the number of bits needed to optimally encode independent draws of a discreet variable Q function parameterized by the current state, action and neural network weights $Q(x, a, \phi)$ following a probability distribution $p(Q(x, a, \phi))$. It is given by the following equation:

$$H = \sum_n p(Q(x, a, \phi)) log(p(Q(x, a, \phi))) \qquad (13)$$

where $i = 1, 2, ..., n$, for all states (total of $n$) the value $Q(x, a, \phi)$ can assume. We let $H_{max}$ denote the maximum entropy of the predicted *Q-value* by the neural network, then intrinsic reward is calculated as follows:

$$H_{max} = n \times p(Q(x, a, \phi))log(p(Q(x, a, \phi))) \quad s.t.$$
$$\tag{14a}$$

$$Q(x, a, \phi) = Q(x, b, \phi) \quad \forall a, b \in \mathcal{A} \tag{14b}$$

$$r^{int} = H_{max} - H \tag{14c}$$

### 3.4. Neural network architecture

In Fig 3, we show our network architecture for the training. The input for the network are single channel images captured using the sawyer head camera or the sawyer arm camera, resized to dimension of 80x80. The resized images go through a series of 2D-Convolution and max pooling layers to capture the features. Next, the output from the third max pooling layer is flattened and fed to 2 dense layers with 512 units and 100 units respectively. A difficult problem in using Q learning in case of large action spaces is that the network has poor convergence properties and often takes an intractable number of samples to train completely. We overcome this problem by decoupling each action and discretizing them into three finer actions (increase, decrease or keep same joint angle). The outputs from the pooling layer is merged with current joint angle inputs and are all connected to a common pooling layer after passing through the convolution layers. Intuitively, all the joint control actions are sharing information from the input image (head/hand) camera, however, the decisions for controlling each joint are decoupled from each other, only depending on the current joint angle configuration.

Thus, for the 7 DoF sawyer arm joint, there are seven separate dedicated sub-networks that take individual joint angle change decisions, but receive the same global reward from the reinforcement learning framework depending on how close the manipulator moved towards the object. Besides the seven sub-networks each mode also contains an eighth similar sub-network that is used to train the switching between the sawyer head camera and the arm camera. Depending on the output of this sub network, the focus of the training is shifted from one state to another. This sub network too receives intrinsic reward whereas the remaining seven networks receive extrinsic reward. The eighth sub-network also uses sigmoid activation function, while the other have linear activation functions. The output of these sub-networks are finally concatenated to a vector that represents the current combined control input (control for current mode and switching decision) to the system. Note that each sub-network for the joints will output 3 values which corresponds to the decisions to increase, decrease or to remain the current joint angle for corresponding joints, while the sub network for switching will output $m = 2$

values denoting probabilities of staying in different modes, based on which switching is executed in the next action.

## 4. Experiment and results

In this section we will explain the experiments that were performed both in the simulation and real setup and the associated results. The neural network function approximator is shown in figure 3. Standard parameters were chosen for the target DQN algorithm as well as the neural network optimizers. Firstly, for the parameters of the Reinforcement Learning (RL) framework, a value of 0.1 is chosen as the target weight update rate, the experience buffer was chosen as a deque data structure with a length of 2000 time points. The discount factor $\gamma$ was chosen to be 0.9, giving a fair weightage to long term actions. An initial value of random exploration parameter $\epsilon$ was chosen to be 0.1, which was subject to exponential decay over epochs. For the neural network, Root Mean Square Propagation (RMSprop) optimization scheme was chosen (Mukkamala & Hein, 2017), with a learning rate of 0.001, $\rho = 0.9$ and decay set to 0. Mean square error was slightly modified to the Huber loss function (Sun et al., 2018) as it provides better results in practice. Rectified Linear Units (ReLu) (Nair & Hinton, 2010) was used as the activation function in the hidden units whereas a linear activation was provided at the output for the final layer in joint control actions and a sigmoid activation was used for the final layer which output probability of switching states. The positions of the object to track were changed using a random normal distribution with three sigma bounds within bounds of the table. The weights of the neural network were optimized for 18000 epochs in the Gazebo simulator setup, after which it was transferred to the real world setup. Thereafter, considerable results were observed on the real setup after training for more than 6000 epochs.

Figure 4 and 5 shows the averaged rewards obtained over the epochs for the real and simulation setups respectively. It is clear that the over time, the algorithm returns improved performance until it saturates. Thereafter on further training overfitting (Tetko et al., 1995) may take place where it puts greater stress on following a single trajectory and fails at tracking the object for multiple positions. To define the performance of the algorithm, we consider the tracking error which is given by:

$$e = \frac{min(c)}{t \times |K|} \tag{15}$$

Here, $min(c)$ is the minimum current distance of the end point of the end effector from the object over an episode and $t$ is the actual distance of the object from the end point of the end effector when the object position was first reset for that episode. $|K|$ denotes the length of the corresponding

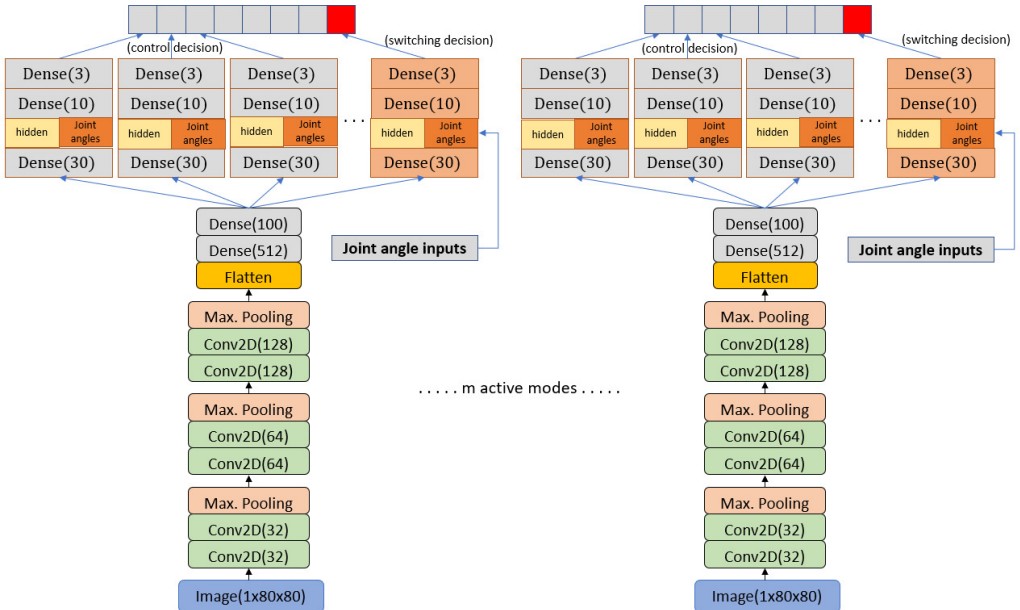

*Figure 3. Neural network architecture*

episode (episode terminates after 50 trials or if the gripper is exactly at the target location). In experiments in figures 7 and 6 it was observed that over epochs the algorithm grows fairly robust to changing object positions as tracking error reduces substantially. However, results in simulator show better performance than real scenario. Finally, figure 2 shows our algorithm in testing scenario for both the simulator and real case. Going from left to right we see that the robot successfully positions the manipulator close to the object (wooden cube in simulator and black cube in real) for two random positions chosen towards left and right of the robot.

## 5. Conclusion

In this paper we presented a control theoretic "state switching" approach as a generic approach in a reinforcement learning setting to solve multi-agent multi input control problem. We reformulated a manipulator control problem for object tracking as a state switching problem and showed that our proposed algorithm performs well in simulation as well as transfers well to the real robot scenario. Our proposed framework relies on original deep Q networks formulation in order to calculate intrinsic reward. However, in future we will consider extending the framework to tackle continuous action spaces. This may include an actor-critic style deep deterministic policy gradients (DDPG) framework where the state switching decision is embedded in the critic itself.

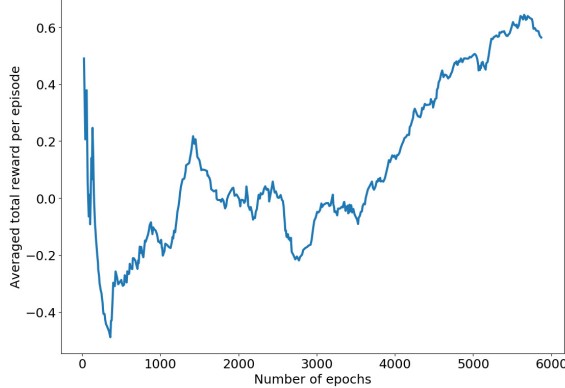

*Figure 4. Averaged cumulative rewards over epochs for real experimental setup- x axis shows the number of epochs in training and y axis shows the averaged reward obtained. The spike drop in initial epochs marks the initial bias caused due to transfer from simulator to real world.*

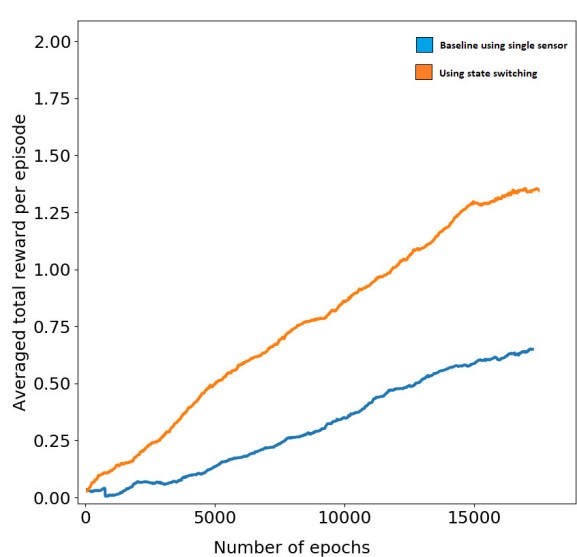

Figure 5. Averaged cumulative rewards over epochs for simulation experimental setup- x axis shows the number of epochs in training and y axis shows the averaged reward obtained. State switching agent (orange) clearly outperforms single mode agents.

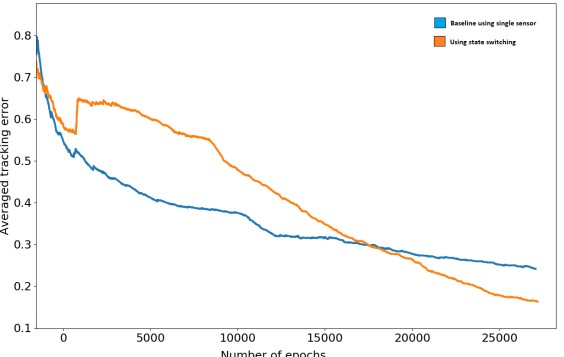

Figure 7. Averaged tracking error over epochs for simulated experimental setup- x axis shows the number of epochs in training and y axis shows the averaged tracking error. State switching causes high variance (orange plot) in the initial epochs, however finally it outperforms single sensor agents.

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

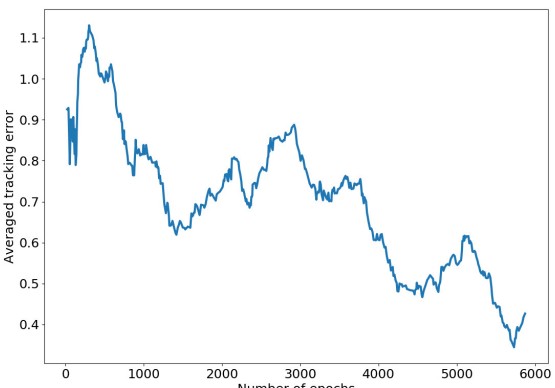

Figure 6. Averaged tracking error over epochs for real experimental setup- x axis shows the number of epochs in training and y axis shows the averaged tracking error.

He, P. and Jagannathan, S. Reinforcement learning-based output feedback control of nonlinear systems with input constraints. *IEEE Transactions on Systems, Man, and Cybernetics, Part B (Cybernetics)*, 35(1):150–154, 2005.

Heydari, A. Optimal scheduling for reference tracking or state regulation using reinforcement learning. *Journal of the Franklin Institute*, 352(8):3285–3303, 2015.

Heydari, A. and Balakrishnan, S. N. Optimal orbit transfer with on-off actuators using a closed form optimal switching scheme. In *AIAA Guidance, Navigation, and Control (GNC) Conference*, pp. 4635, 2013a.

Heydari, A. and Balakrishnan, S. N. Optimal multi-therapeutic hiv treatment using a global optimal switching scheme. *Applied Mathematics and Computation*, 219(14): 7872–7881, 2013b.

Heydari, A. and Balakrishnan, S. N. Optimal switching and control of nonlinear switching systems using approximate dynamic programming. *IEEE Transactions on Neural Networks and Learning Systems*, 25(6):1106–1117, 2014.

Howard, R. A. Dynamic programming and markov processes. 1960.

Kahn, G., Sujan, P., Patil, S., Bopardikar, S., Ryde, J., Goldberg, K., and Abbeel, P. Active exploration using trajectory optimization for robotic grasping in the presence of occlusions. In *2015 IEEE International Conference on Robotics and Automation (ICRA)*, pp. 4783–4790. IEEE, 2015.

Kamgarpour, M. and Tomlin, C. On optimal control of non-autonomous switched systems with a fixed mode sequence. *Automatica*, 48(6):1177–1181, 2012.

Krainin, M., Henry, P., Ren, X., and Fox, D. Manipulator and object tracking for in hand model acquisition. In *Proceedings, IEEE International Conference on Robots and Automation*, pp. 1817–1824, 2010.

Liu, C. and Gong, Z. Modeling and optimal control of a time-delayed switched system in fed-batch process. *Journal of the Franklin Institute*, 351(2):840–856, 2014.

Luus, R. and Chen, Y. Optimal switching control via direct search optimization. In *Proceedings of the 2003 IEEE International Symposium on Intelligent Control*, pp. 371–376. IEEE, 2003.

Mnih, V., Kavukcuoglu, K., Silver, D., Graves, A., Antonoglou, I., Wierstra, D., and Riedmiller, M. Playing atari with deep reinforcement learning. *arXiv preprint arXiv:1312.5602*, 2013.

Mukkamala, M. C. and Hein, M. Variants of rmsprop and adagrad with logarithmic regret bounds. In *Proceedings of the 34th International Conference on Machine Learning-Volume 70*, pp. 2545–2553. JMLR. org, 2017.

Nair, V. and Hinton, G. E. Rectified linear units improve restricted boltzmann machines. In *Proceedings of the 27th international conference on machine learning (ICML-10)*, pp. 807–814, 2010.

Rinehart, M., Dahleh, M., Reed, D., and Kolmanovsky, I. Suboptimal control of switched systems with an application to the disc engine. *IEEE Transactions on Control Systems Technology*, 16(2):189–201, 2008.

Rungger, M. and Stursberg, O. A numerical method for hybrid optimal control based on dynamic programming. *Nonlinear Analysis: Hybrid Systems*, 5(2):254–274, 2011.

Sun, Q., Zhou, W.-X., and Fan, J. Adaptive huber regression. *Journal of the American Statistical Association*, (just-accepted):1–35, 2018.

Tamar, A., Wu, Y., Thomas, G., Levine, S., and Abbeel, P. Value iteration networks. In *Advances in Neural Information Processing Systems*, pp. 2154–2162, 2016.

Tetko, I. V., Livingstone, D. J., and Luik, A. I. Neural network studies. 1. comparison of overfitting and overtraining. *Journal of chemical information and computer sciences*, 35(5):826–833, 1995.

Wang, D., Jia, W., Yu, Y., and Wang, W. Recognition and grasping of target position and pose of manipulator based on vision. In *2018 5th International Conference on Information, Cybernetics, and Computational Social Systems (ICCSS)*, pp. 483–488. IEEE, 2018.

Wardi, Y. and Egerstedt, M. Algorithm for optimal mode scheduling in switched systems. In *2012 American Control Conference (ACC)*, pp. 4546–4551. IEEE, 2012.

Wunder, M., Littman, M. L., and Babes, M. Classes of multiagent q-learning dynamics with epsilon-greedy exploration. In *Proceedings of the 27th International Conference on Machine Learning (ICML-10)*, pp. 1167–1174. Citeseer, 2010.

Xu, X. and Antsaklis, P. J. Optimal control of switched systems via non-linear optimization based on direct differentiations of value functions. *International Journal of Control*, 75(16-17):1406–1426, 2002.

Xu, X. and Antsaklis, P. J. Optimal control of switched systems based on parameterization of the switching instants. *IEEE transactions on automatic control*, 49(1): 2–16, 2004.
