# OpenReview forum: "Learning State Switching for Improved Exploration in Multi-sensor Environments"
_ICML.cc/2019/Workshop/RL4RealLife — Submitted to RL4RealLife 2019_

### Official Review · AnonReviewer1 · 2019-05-26
**An interesting application of reinforcement learning but with weak execution**

**Rating:** 2
**Confidence:** 4

**Review:**

The paper applies reinforcement learning, in particular DQN, to a multi-dimensional controller. One interesting aspect of the application is that the agent needs to decide on both the control input as well as the mode(state)-switch so the agent can take into account of different sensor input at each timestep. The authors demonstrate the performance of the agent in simulated and real environment and show using state-switching improves cumulative reward.

The notation of the paper is rather confusing. Section 2 is written with notations specific to the control problem, while section 3 is written in notations common in the DRL literature without specifying the correspondence.

The main novelty of work, Algorithm 1, is the action space include both the state switching  as well as control input. Not enough clarity or emphasis was put on that in section 3 though. It reads as if an out-of-the-box application of DQN without much clarification of the challenges faced or modifications required to make it work.

While authors do compare baseline agent with single sensor to the multi-sensor with state-switching and show improvement. There is no comparison to any other baselines or alternatives for similar problems provided.

---

### Decision · Program_Chairs · 2019-05-28

Reject